



# IAQMS-street v2.0: a two-way coupled regional-urban–street-network model system for Beijing, China

Tao Wang[1,2], Hang Liu[1,2], Jie Li[1,2], Shuai Wang[3], Youngseob Kim[4], Yele Sun[1], Wenyi Yang[1], Huiyun Du[1], Zhe Wang[1], Zifa Wang[1,2]

[1]LAPC, Institute of Atmospheric Physics, Chinese Academy of Sciences, Beijing, China
[2]University of Chinese Academy of Sciences, Beijing, China
[3]China National Environmental Monitoring Centre, Beijing, China
[4]CEREA, École des Ponts , EDF R&D, Marne-la-Vallée, France

*Correspondence to*: Jie Li, Zifa Wang (lijie8074@mail.iap.ac.cn; zifawang@mail.iap.ac.cn)

**Abstract.** Owing to the substantial traffic emissions in urban areas, especially near road areas, the concentrations of pollutants, such as ozone ($O_3$) and its precursors, have a large gap with the regional averages and their distributions cannot be captured accurately by traditional single-scale air-quality models. In this study, a new version of a regional-urban-street-network model (IAQMS-street v2.0) is presented. An upscaling module is implemented in IAQMS-street v2.0 to calculate the impact of mass transfer to regional scale from street network. The influence of pollutants in street network is considered in the concentration calculation on regional scale, which is not considered in a previous version (IAQMS-street v1.0). In this study, the simulated results in Beijing during August 2021 by using IAQMS-street v2.0, IAQMS-street v1.0, and the regional model (NAQPMS) are compared. On-road traffic emissions in Beijing, as the key model-input data, were established using intelligent image-recognition technology and real-time traffic big data from navigation applications. The simulated results showed that the $O_3$ and nitrogen oxides ($NO_x$) concentrations in Beijing were reproduced by using IAQMS-street v2.0 both on regional scale and street scale. The prediction fractions within a factor of two (FAC2s) between simulations and observations of NO and $NO_2$ increased from 0.11 and 0.34 in NAQPMS to 0.78 and 1.00 in IAQMS-street v2.0, respectively. The normalized mean biases (NMBs) of NO and $NO_2$ decreased from 2.67 and 1.33 to -0.25 and 0.08. the concentration of $NO_x$ at street scale is higher than that at the regional scale, and the simulated distribution of pollutants on regional scale was improved in IAQMS-street v2.0 compared with that in IAQMS-street v1.0. We further used the IAQMS-street v2.0 to quantify the contribution of local on-road traffic emissions to the $O_3$ and $NO_x$ emissions and analyze the effect of traffic-regulation policies in Beijing. Results showed that heavy-duty trucks are the major source of on-road traffic emissions of $NO_x$. The relative contributions of local traffic emissions to $NO_2$, NO, and $O_3$ emissions were 53.41, 57.45, and 8.49%, respectively. We found that traffic-regulation policies in Beijing largely decreased the concentrations of $NO_x$ and hydrocarbons (HC); however, the $O_3$ concentration near the road increased due to the decrease consumption of $O_3$ by NO. To decrease the $O_3$ concentration in urban areas, controlling the local emissions of HC and $NO_x$ from other sources requires consideration.

**Short summary.** This paper developed a two-way coupled module in latest version of a regional-urban-street-network model IAQMS-street v2.0, the mass flux from streets to background is considered. Test cases are defined to evaluate the performance of IAQMS-street v2.0 in Beijing by comparing it with that simulated by IAQMS-street v1.0 and regional

10.5194/gmd-2023-5





the polluted days in 2019 were caused by $O_3$ (Ministry of the Ecological Environment of China, 2020a). Owing to its
increasing concentration trend and adverse impacts on humans and vegetation, an increasing number of studies have focused
on the mechanism of $O_3$ formation as well as relevant control strategies.

The $O_3$ concentration is influenced by the meteorological fields, precursor-emission intensities, photochemical processes,
and regional transport processes (Zheng et al., 2018; Wang et al., 2017a). As the precursors of $O_3$, nitrogen oxides ($NO_x$) and
volatile organic compounds (VOCs) have complicated nonlinear relationships. The formation of surface $O_3$ can be divided
into the $NO_x$-sensitive and VOC-sensitive regions, owing to the complexity of photochemical processes (Sillman, 1999),
with the primary control species of the precursors requiring careful consideration according to the sensitive region in each
case. The $O_3$ concentration may even increase after conducting inappropriate precursor control; consequently, increasing the
precision of simulations of $O_3$ and its precursors at the urban scale constitutes an urgent scientific topic.

Regional-scale air-quality models are common tools for analyzing air pollution episodes, such as Comprehensive Air Quality
Model with Extensions (CAMx), the Community Multi-scale Air Quality (CMAQ) model (Byun and Schere, 2006),  and
Nested Air Quality Prediction Modeling System (NAQPMS) have been widely used in air-quality research (Li et al., 2012a;
Wang et al., 2017b; Cheng et al., 2019; Zhang et al., 2020a). The influence of anthropogenic emissions on the regional
atmospheric environment has been assessed through sensitivity analyses using regional models (Zhang et al., 2020a; Cheng
et al., 2019) or source-apportionment analyses  (Wagstrom et al., 2008; Yarwood et al., 1996; Wang et al., 2017b; Lin et al.,
2016; Li et al., 2015; Li et al., 2012b). However, regional models have spatial resolutions that are usually coarser than $1 \times 1$
km, thereby being unable to capture the emission and diffusion characteristics of pollutants at the street scale (Baik and Kim,
2010). Thus, the influence of local emissions on air quality at the street scale cannot be simulated using regional models.

Local-scale air-quality models, such as the computational fluid dynamic (CFD) and street-scale network models, which
consider the impact of urban building topography on the diffusion of pollutants (Depaul and Sheih, 1985, 1986; Wedding et
al., 1977), have been adopted by numerous researchers to investigate the distribution of pollutants at a finer spatial resolution
(Vardoulakis et al., 2003; Zhang et al., 2020b; Patterson and Harley, 2019; Soulhac et al., 2012). The flow field and
dispersion of pollutants in local scale such as street canyons can be accurately simulated by the CFD model, but it is suitable
for air-quality simulations over a few streets rather than at the urban scale. In addition, the CFD model does not usually
consider complex chemical reactions; this introduces limitations to the simulation of secondary pollutants, such as $O_3$
(Fellini et al., 2019; Thouron et al., 2019; Ashie and Kono, 2011). Street-scale network models, such as the Model of Urban
Network of Intersecting Canyons and Highways (MUNICH),  operational urban dispersion model (SIRANE), and the
Operational Street Pollution Model (OSPM) (Kakosimos et al., 2010; Soulhac et al., 2011; Kim et al., 2018; Kim et al.,
2022) can simulate the distribution of pollutants at the street scale with a lower computational cost. MUNICH has been
widely used for investigating the air quality at the street scale (Gavidia-Calderón et al., 2021; Lugon et al., 2020; Kim et al.,
2018). Further studies showed that the simulations of the street-scale model are influenced by the utilized background field
in each case, which may be provided by a regional model (Lv et al., 2022; Wang et al., 2022; Kim et al., 2018). Therefore, to
provide a more dynamic and precise background field, it is essential to build a two-way integrated regional-urban–street-
network air-quality model, the feedback of street-network scale model on regional-urban background needs to be considered
Many researchers have focused on the development and application of coupled regional-urban-street-network scale air-
quality models (Lv et al., 2022; Nuterman et al., 2021; Biggart et al., 2020; Lugon et al., 2020; Benavides et al., 2019; Kim
et al., 2018; Hood et al., 2018; Isakov et al., 2007; Isakov et al., 2009). However, most coupled models involve large
uncertainties originating mainly from traffic emissions; hence, they need more refined emission inventories as input (Biggart
et al., 2020).

The variation of the emission inventory of $O_3$ precursors is critical to $O_3$ generation. Traffic emissions become one of the
main sources of $O_3$ precursors, especially in urban areas. Cheng et al. (2019) found that the anthropogenic emissions of $NO_x$
and VOCs in 2017 decreased by 42.9 and 42.4%, respectively, compared with the emissions in 2013, owing to strict
industrial emission control in China. However, the contribution of traffic to the $NO_x$ emissions increased from 67.2% in
2013 to >80% in 2017.  In additional, emission uncertainties, caused by the spatial mismatch between the locations of
emissions and spatial proxies, can lead to additional uncertainties in air-quality simulations, especially in small-scale regions





(Zheng et al., 2017); hence, real-time high-resolution traffic-emission inventories are also essential for more precise coupled-model simulations.

In this study, we developed a new version of dynamic urban-street scale model (IAQMS-street v2.0) using a two-way coupling between the MUNICH street-scale model and NAQPMS regional air-quality model based on previous version of IAQMS-street v1.0 (Wang et al., 2022), the hourly variation and spatial distribution of $O_3$ and $NO_x$ concentrations  were

simulated in Beijing during August 2021. An upscaling module is added to transfer the pollutants from street scale model to regional model, which is not considered in IAQMS-street v1.0. We used image-recognition technology based on road monitoring and traffic big data to create high-resolution traffic-emission inventories. To evaluate the performance of the IAQMS-street v2.0, we conducted simulations under different models (IAQMS-street v2.0, IAQMS-stree v1.0, and NAQPMS) and validated the simulation results through comparison with observations from monitoring sites as well as on-

road observations. In the following, we discuss the simulation differences among the two-way coupled (IAQMS-street v2.0), one-way coupled (IAQMS-street v1.0), and regional models (NAQPMS), and analyze the $O_3$ and $NO_x$ distribution characteristics. Furthermore, we quantify the contribution of on-road vehicle emissions to the distribution of $O_3$ and $NO_x$ concentration. The influence of traffic management and control measures on the variation of traffic emissions and pollutant concentrations are quantified.

**2 Materials and methods**

**2.1 Coupled regional-urban–street-network scale model**

As the regional scale model used in IAQMS-street v2.0, the NAQPMS regional air-quality model is a 3-dimensional Eulerian chemical transport model, reproduce the chemical and physical process of pollutants by solving the mass balance equations The physical processes include the horizontal/vertical advection and diffusion, dry/wet deposition process in

NAQPMS, and it includes also a gaseous chemical mechanism (CBM-Z) for chemical reaction processes of pollutants. NAQPMS has been widely used for investigating regional pollution events in China (Yang et al., 2019; Wang et al., 2014a; Wang et al., 2014b; Lin et al., 2007; Wang et al., 2006), since it performs well in operational forecasting. For additional information on the regional model, we refer to Li et al. (2007), (2011), and (2012a).

The pollutants concentration in NAQMS at next time step is calculated as follows:

$$C_{t+dt} = C_t + dC, \tag{1}$$

$$dc = C_{emiss} + C_{adv} + C_{diff} + C_{chem} - C_{dep}, \tag{2}$$

Where dt is the time step, $C_t$ is the concentration in grid cell at time t, $C_{t+dt}$ is the concentration at next time. As show in Eq. (2), the variation of pollutants concentration dc is influenced by emissions process ($C_{emiss}$), advection process ($C_{adv}$), diffusion process ($C_{diff}$), chemical process ($C_{chem}$), and dry/wet deposition process ($C_{dep}$).

The MUNICH street-network model was developed to simulate the concentration of pollutants in street-network by Kim et al. (2018). The emission, dry/wet deposition, horizontal transport, and vertical transport process between the background and urban canopy were included in the model; it includes also a gaseous chemical mechanism (CB05), whose species were matched with those of CBM-Z in the regional model during the coupling process. For a more detailed description of MUNICH, we refer to Kim et al. (2018) and Lugon et al. (2020).

In MUNICH, the pollutants concentration in streets are calculated as follows:

$$C_{street} = \frac{Q_{emis} + Q_{inflow} + \gamma C_{bg}}{\gamma + Q_{vert} + F_{dep}}, \tag{3}$$

Where both the chemical process and physical process are considered in the calculation of pollutants concentration, and the physical process included the inflow rate of pollutant between streets ($Q_{inflow}$), the vertical transfer process between streets and urban background atmosphere ($Q_{vert}$), the traffic emission rate from on-road vehicle ($Q_{emis}$), and dry/wet deposition

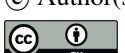



process ($F_{dep}$). $C_{bg}$ is the background concentration which simulated by regional model. $\gamma$ is related to mass flux between street and background concentration :

$$Q_{vert} = \gamma(C_{street} - C_{bg}),\qquad(4)$$

The introduction of detail parameter settings in MUNICH can be found in Lugon et al. (2020).

An one-way integrated air-quality modeling system named IAQMS-street v1.0 have been developed in previous research(Wang et al., 2022), In IAQMS-street v1.0, the pollutants background concentrations in study period were simulated by NAQPMS, and as the input data of MUNICH, the background concentration were provided for the simulation of pollutants at street scale. MUNICH was used as a standalone model with a one-way coupling approach. The influence of mass transfer of pollutants from streets to urban background was not considered in IAQMS-street v1.0. In this study, the IAQMS-street was further developed by adding an upscaling module to achieve the feedback of MUNICH to NAQPMS (named IAQMS-street v2.0), thereby influencing the variation in background concentrations. The simulation results from NAQPMS and MUNICH were two-way coupled to represent the hourly variation and spatial distribution of air pollutants at the regional scale and local scale.

In IAQMS-street v2.0, the influence of mass flux from street in MUNICH to grid cell in NAQPMS are considered to update the background concentration. An upscaling module is added in IAQMS-street v2.0: the pollutants were transferred from street to urban background by $Q_{vert}$ in Eq. (3), the mass flux of pollutants is setting as an added traffic emission $C_{emiss}$ in Eq. (2) to calculate the concentration in grid cell at next time step. The pollutants concentration in grid cell at the next time step were calculated by pollutants mass in street and background mass in grid cell.

$$C_{grid} = \frac{M_{grid}}{V_{grid}} = \frac{M_{bg}+M_{street}}{V_{grid}},\qquad(5)$$

Where $C_{grid}$ is the mean pollutants concentration in grid cell, $M_{street}$ is the pollutant mass in streets, $M_{bg}$ is the background pollutant mass, $V_{grid}$ is grid volume which include street volume. In this study, NAQPMS and MUNICH were two-way coupled and applied to Beijing. The coupling schematic diagram of NAQPMS and MUNICH is show in Fig. 1. The simulation results of MUNICH at street scale are related to the simulated concentration at bottom layer of NAQPMS. In the two-way coupled module, the background concentration and meteorological data were simulated by NAQPMS and provided to street scale by downscaling module, and the influence of mass flux of pollutant from street to regional background were considered to NAQPMS by upscaling module. MUNICH was located within the lowest NAQPMS layer.. After the calculation of the mass flux between the urban canopy and urban background, the upscaling module would transfer the pollutants from MUNICH to NAQPMS to compute the polllutant concentrations in bottom layer of NAQPMS. The fixed time step for interfacing between NAQPMS and MUNICH was 20 min, i.e., the same as that of the regional model.





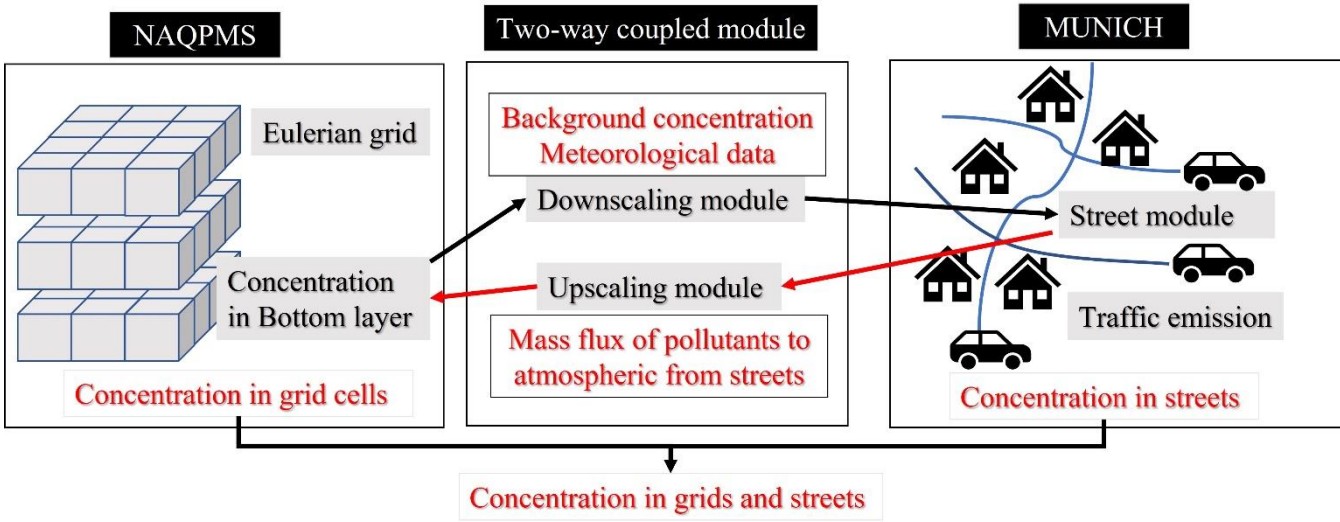

165

**Figure 1.** The framework of the two-way coupled model IAQMS-street v2.0. An upscaling module is added in the two-way coupled module to transfer the calculated mass flux between streets and regional background in the street-network model to the regional model.

The simulation space range is a two-level nested domains in NAQPMS (as shown in Fig. 2a), with the largest domain (d01)
170 covering the middle and east of China, the horizontal resolution of d01 in NAQPMS is 9 km. In inner domain (d02), the simulation space range covering the whole Beijing area, the horizontal resolution of d02 in this study is 1 km. The domain setting covered the Beijing area in MUNICH, the streets location and observation sites in Beijing are shown in Fig. 2b. The simulation of the surface $O_3$ and $NO_x$ concentrations was conducted from 1 to 31 August 2021, when the photochemical reactions were strong.

175



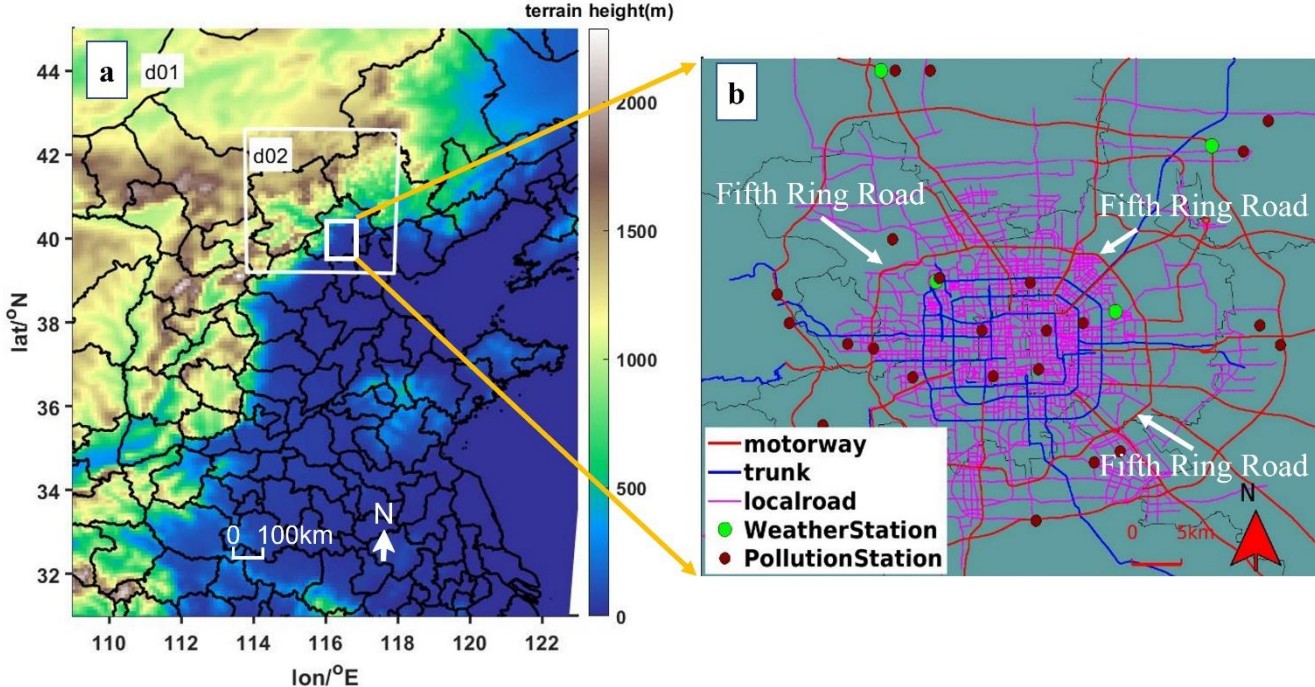

**Figure 2.** (a). Simulation domains at regional scale: the largest domain 1 (d01) covers the middle and east of China with a horizontal grid spacing of 9 km, an inner domain (d02) covers Beijing and surround areas with a horizontal grid spacing of 1 km. (b). The modelling area and street network in the street-scale model. Green and brown points indicate the locations of urban monitoring stations.

## 2.2 Traffic emission model

As essential inputs to regional air quality model, emission inventories are important to the simulation results. The additional uncertainties of simulation results arose for grided emissions with finer resolutions because of spatial errors, especially in urban areas (Zheng et al., 2017). In this study, on-road traffic emissions were calculated based on real-time traffic speed data and road-vehicle recognition technology to reduce additional uncertainties.

To obtain dynamic high-resolution traffic emission inventories, a real-time on-road traffic emission model (ROE) was developed by Wu et al. (2020),  the street network traffic emissions was calculated by using real-time traffic-speed data, traffic volume, and vehicle emission factors.. Navigation application such as Gaode Map and Baidu Map provide the original traffic-speed data. Based on traffic big data, the traffic volume was calculated by using the Underwood speed-volume calculation formula (Greenshields et al., 1961), with the proportion of different vehicles on the road being set before using the ROE. For a detailed description of ROE, we refer to Wu et al. (2020); for the model configuration of traffic emissions in Beijing, we refer to Wang et al. (2022).

It is worth noting that because of the implementation of road-traffic control measures (e.g., diesel vehicles below the National Grade IV are forbidden from entering the Fifth Ring Road in Beijing), the proportions of vehicles in urban and suburban areas were different in Beijing. To obtain information on road vehicles on urban and suburban roads in Beijing, the traffic flow and proportion of vehicles on different roads were counted using a real-time object detection system (YOLOv5s; https://github.com/ultralytics/yolov5; last access: September 5th, 2022). Sampling locations for vehicles at motorways, trunks, and local roads in Beijing during the study period are shown in Fig. 3a; the road-vehicle detection results are shown in Fig. 3b.





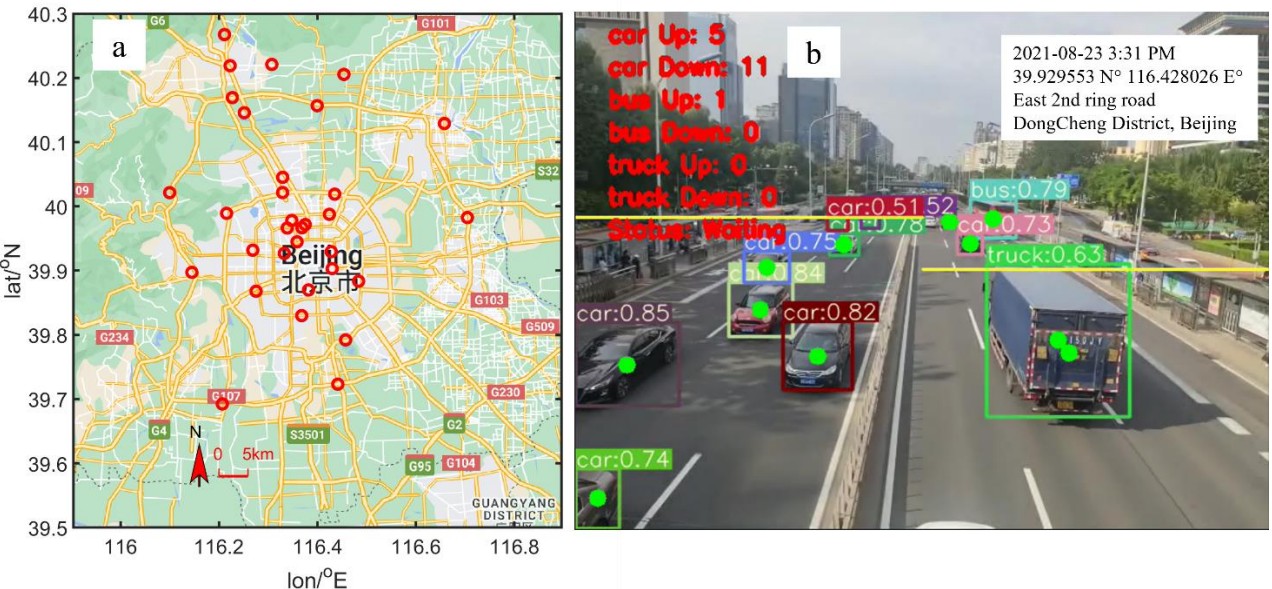

**Figure 3.** (a) Locations of observation sites on different roads for vehicle information (Imagery © 2022 Google, map data © 2022 Google). (b) Detection results of vehicles on road by the YOLO system.

### 2.3 Simulation scenarios

Model performances were evaluated through five simulation scenarios (Table 1). Three scenarios with different models, involving a two-way coupled model (S1: IAQMS-street v2.0), one-way coupled model (S2: IAQMS-street v1.0), and regional model (S3: NAQPMS), were simulated with hourly dynamic traffic emissions generated by the ROE model. The simulated $O_3$ and $NO_x$ concentrations in these three models were compared with observational data to demonstrate the differences between the coupled and regional models. In addition, we established two other scenarios (based on S1, albeit with different emissions) to investigate the impact of different traffic-control measures. We assumed that the proportion of vehicles in urban areas within the Fifth Ring Road was the same as the proportion of vehicles in suburban areas in the scenario without a low-emission zone (i.e., S1_withoutLEZ). The simulation was compared with S1 to evaluate the impact of low-emission zones on air quality. All vehicles on roads in Beijing, include petrol and diesel vehicles, were assumed to meet the National V emission standards in the scenario with upgraded traffic emission standards (S1_upgrade), and the impact of the upgraded emission standard on pollution concentration was evaluated by comparison with S1. These models and scenarios were used to simulate the concentration variations of $O_3$ and its precursors in August 2021. The period from 20 to 31 July was in each case the spin-up period.

**Table 1.** List of the simulations performed by two-way coupled model (IAQMS-street v2.0), one-way coupled model (IAQMS-street v1.0), and regional model (NAQPMS) in this study.

| Scenarios | Model | Simulated range (Horizontal resolution) | Traffic emission |
|---|---|---|---|
| S1 | IAQMS-street v2.0 | Urban (1 km)/Street (100 m) | Dynamic emission |
| S2 | IAQMS-street v1.0 | Urban (1 km)/Street (100 m) | Dynamic emission |
| S3 | NAQPMS | Urban (1 km) | Dynamic emission |
| S1_withoutLEZ | IAQMS-street v2.0 | Urban (1 km)/Street (100 m) | Without low emission zone |





| S1_upgrade | IAQMS-street v2.0 | Urban (1 km)/Street (100 m) | Upgraded emission standard |

## 3 Results and discussion

### 3.1 On-road vehicle emissions

We used the ROE model to calculate the hourly on-road emission variations of different types of vehicles, including taxi, bus, heavy-duty truck (HDT), middle-duty truck (MDT), light-duty truck (LDT), heavy-duty vehicle (HDV), middle-duty vehicle (MDV), and light-duty vehicle (LDV). The diurnal variations of the on-road traffic emissions in Beijing in August 2021 is shown in Fig. 4. There are generally two high-emission peak hours (i.e., 08:00–10:00 and 18:00–20:00 h LT) on the roads in Beijing, correspond to the commuting times. The HDT contribution to vehicle $NO_x$ emissions reached 34.4%, while vehicle hydrocarbons (HC) emissions mainly originated from the LDVs, reaching 56.4%. The HDT contribution to vehicle $NO_x$ emissions increased to 51.1% at midnight due to the reduced number of private petrol vehicles and the increased nighttime HDT traffic across urban Beijing.

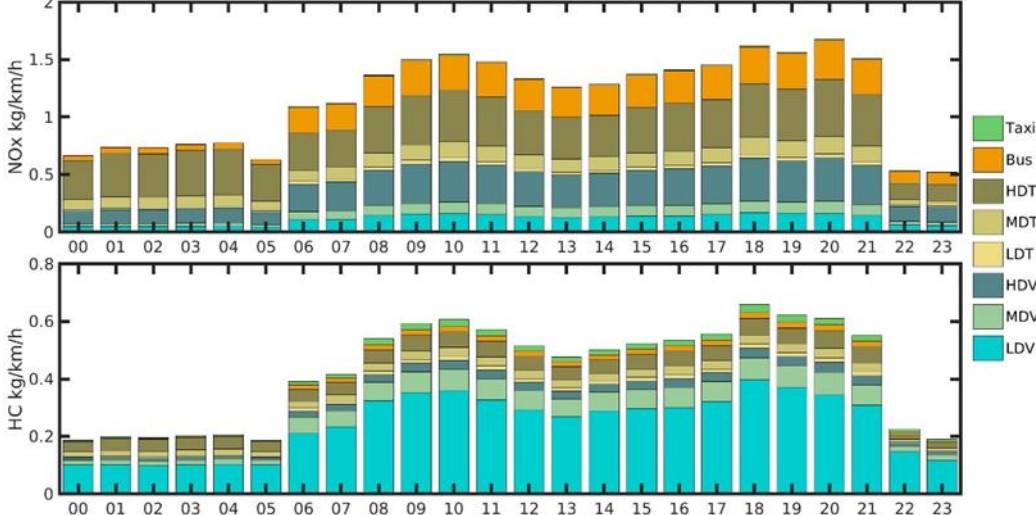

**Figure 4.** Diurnal variations of the contributions of road vehicles to the $NO_x$ emissions and Hydrocarbon (HC) emissions in Beijing during August 2021.

The spatial distributions of the on-road $NO_x$ and HC traffic emissions in Beijing are shown in Fig. 5. For a given area, emissions on ring roads were higher than those on local roads within the Fifth Ring Road because traffic flow on ring roads was higher than that on local roads. In additional, the traffic-control measures and low-emission zones decreased the LDT proportions in urban areas, thereby leading to lower $NO_x$ emissions on roads within the Fifth Ring Road. $NO_x$ emissions intensities exceeding 60 kg/km/day on the Fifth Ring Road and the highway outside the Fifth Ring Road. The $NO_x$ emissions inside the Fifth Ring Road were lower than 20 kg/km/day. HC emissions exhibited a similar spatial distribution; however, the HC emission difference between urban and suburban areas was decreased compared with that of $NO_x$ because HC was mainly emitted by LDVs.

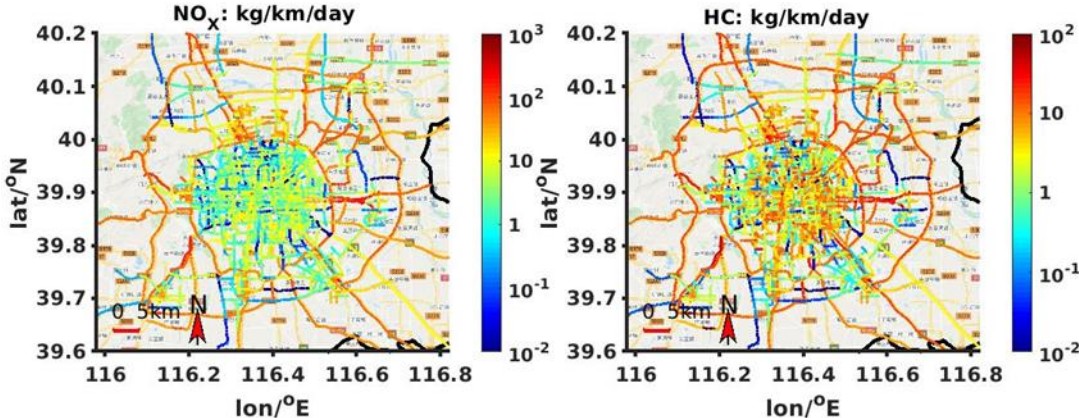

**Figure 5.** Horizontal distributions of the $NO_x$ and HC emissions (unit: kg/km/day) at the street-network in Beijing urban and suburban area during August 2021 (Imagery © 2022 Google, map data © 2022 Google).

### 3.2 Evaluation of the pollutant simulations on the regional and street scales

The simulated $O_3$ and $NO_x$ concentrations by IAQMS-street v2.0, IAQMS-street v1.0, and NAQPMS were evaluated by
comparing with observations from the pollutant monitoring stations during the study period. The results showed that the variation of $O_3$ and $NO_x$ concentrations simulated by three models were consistent with observations. However, the concentrations of pollutants simulated by IAQMS-street v2.0 were closer to the observations compared to those simulated by IAQMS-street v1.0 and NAQPMS, especially on hourly variation of NO and $NO_2$. The nighttime NO and $NO_2$ concentrations were overestimated in NAQPMS, whereas the simulation of IAQMS-street v2.0 was closer to the
observations. The NO and $NO_2$ simulation performance was improved at different stations, especially at midnight, in IAQMS-street v2.0, because the street-scale model was coupled with regional model in the system and the feedback of street scale model on regional model was considered, IAQMS-street v2.0 was more suitable for simulating pollutants in urban areas. The underestimation of $O_3$ during nighttime was improved in IAQMS-street v2.0 compared with that in NAQPMS, because of the weakened $O_3$ depletion by the $NO_x$-$O_3$ titration reaction in IAQMS-street v2.0. Table S1 showed the statistical
parameters for the DongCheng (DC) and XiCheng (XC) district stations , the simulated mean NO concentration during August 2021 decreased from 3.73 (4.12) μg/m$^3$ in NAQPMS to 0.80 (0.77) μg/m$^3$ in IAQMS-street v2.0 at the DC (XC) station, thereby being closer to the observed mean value of 0.57 (0.64) μg/m$^3$. The root-mean-square errors (RMSEs) of the NO, $NO_2$, and $O_3$ concentrations decreased from 5.96–6.60, 27.00–27.31, and 45.64–54.20 μg/m$^3$ in NAQPMS to 1.53–1.71, 14.21–14.82, and 30.18–33.42 μg/m$^3$ in S1, respectively. Overall, the simulation results of IAQMS-street v2.0 were closer to
the observations; in IAQMS-street v1.0, the NO and $O_3$ concentrations were underestimated and the $NO_2$ concentration was overestimated at the street scale owing to the lack of feedback from the street urban canopy to the regional-urban background.



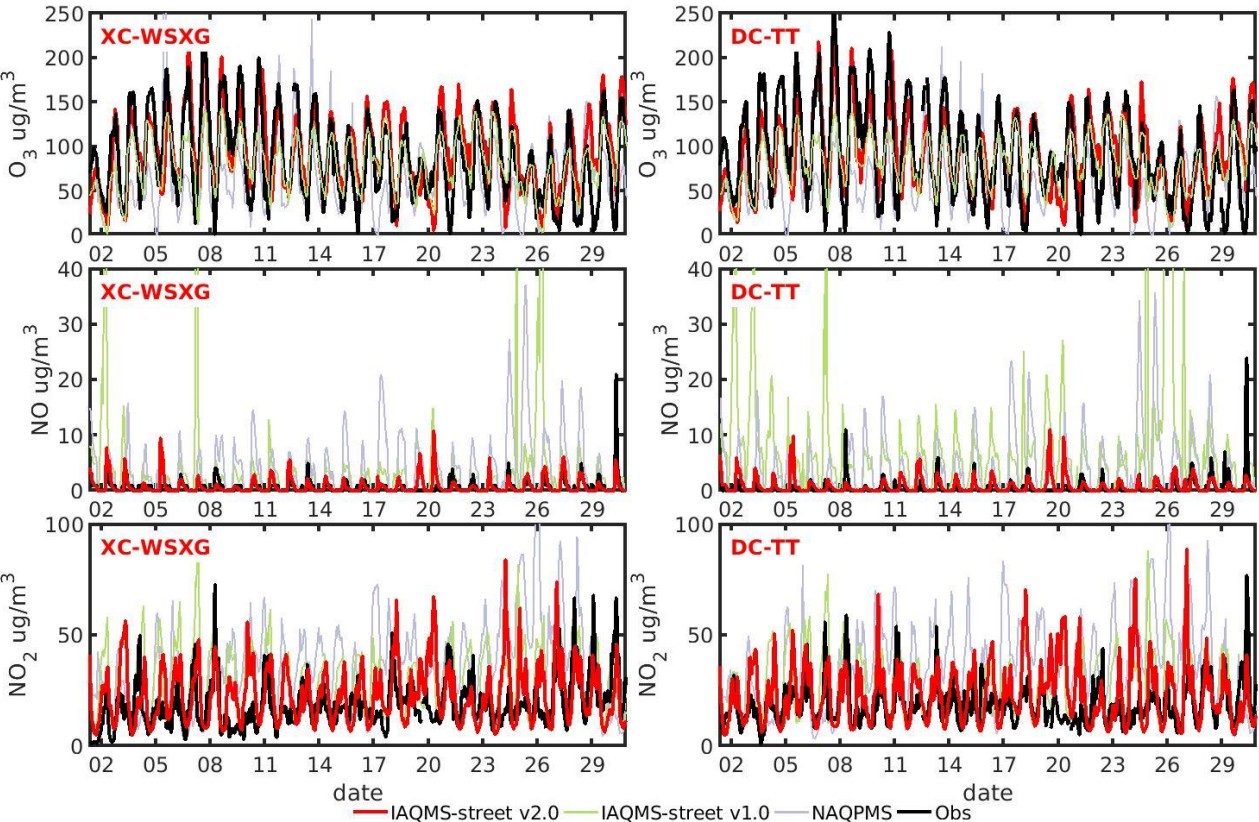

**Figure 6.** Hourly variation of $O_3$, NO, and $NO_2$ concentrations (unit: $\mu g/m^3$) during August 2021 at the Dongcheng-Tiantan (DC-TT) station and Xicheng-Wanshouxigong (XC-WSXG) station. Red lines indicate values simulated by IAQMS-street v2.0; green lines indicate values simulated by the IAQMS-street v1.0; blue lines indicate values simulated by NAQPMS; and black lines indicate observations.

The observed and simulated mean $O_3$, $NO_x$ concentrations during August 2021 in IAQMS-street v2.0, IAQMS-street v1.0, and NAQPMS at all pollutant monitoring stations (site information is shown in Fig. 2b) in Beijing are shown in Fig. 7. The simulation results of the two-way coupled model (IAQMS-street v2.0) were improved by comparing with those of the one-way coupled model (IAQMS-street v1.0) and the regional model (NAQPMS). The prediction fractions within a factor of two (FAC2s) between simulations and observations of NO and $NO_2$ increased from 0.11 and 0.34 in NAQPMS to 0.78 and 1.00 in IAQMS-street v2.0, respectively. The normalized mean biases (NMBs) of NO and $NO_2$ decreased from 2.67 and 1.33 in NAQPMS to -0.25 and 0.08, respectively.



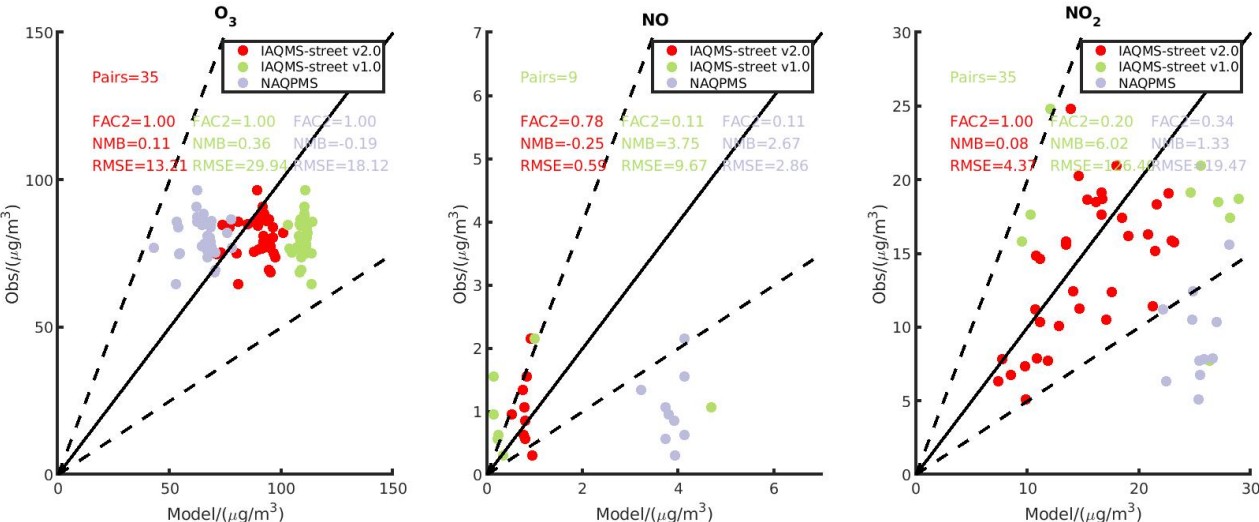

**Figure 7.** Observed and simulated average $O_3$, NO, and $NO_2$ concentrations during August 2021from different models (IAQMS-street v2.0: blue points; IAQMS-street v1.0: red points; NAQPMS: black points) at all pollutant monitoring stations in Beijing.

The spatial distributions of the simulated monthly averaged $O_3$ and $NO_x$ concentrations during August, 2021 in three models
are shown in Fig. 8. At the urban scale, high $NO_x$ concentrations appeared near road areas, especially on busy roads. $O_3$ exhibited opposite concentration trends in IAQMS-street v2.0, meaning that it reacted due to the high NO concentration near roads. The $NO_x$ concentrations near highways in suburban areas were higher than those within the Fifth Ring Road because of the implementation of traffic-control measures. The $NO_x$ and $O_3$ distributions exhibited similar trends in IAQMS-street v1.0 compared with IAQMS-street v2.0 at the street scale; however, the $NO_x$ concentration was lower at the regional scale
because local road emissions were not considered in regional model and the influence of mass transfer from the street-scale model were not considered in the regional model. Pollutant distributions in NAQPMS were similar to those in IAQMS-street v2.0; however, the $NO_x$ concentration was higher near the street area because street-scale processes were not accounted by the regional model. The population-weighted average $O_3$, NO, and $NO_2$ concentrations were calculated based on the population-density distribution in Beijing, and the results are shown in Table S2. We found that the population-weighted
average NO and $NO_2$ concentrations increased from 0.12 and 3.75 μg/m$^3$ in IAQMS-street v1.0 to 0.79 and 16.64 μg/m$^3$ in IAQMS-street v2.0, respectively, due to the influence of the street-scale model on the regional model. Moreover, the population-weighted average NO and $NO_2$ concentrations reached 1.50 and 20.13 μg/m$^3$ in NAQPMS because the $NO_x$ concentrations were overestimated at near-road environments.



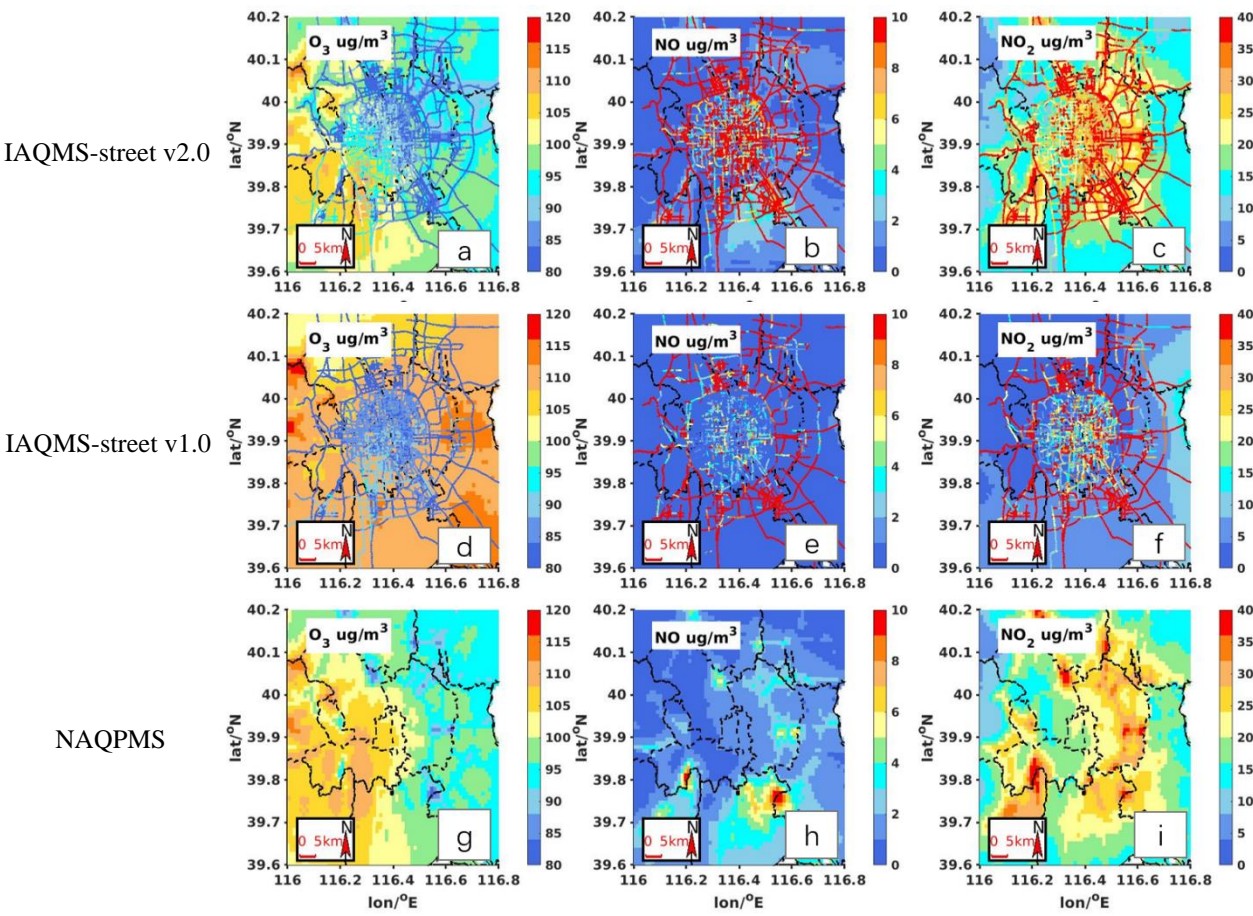

**Figure 8.** Spatial distribution of the monthly averaged $O_3$, NO, and $NO_2$ concentrations (unit: μg/m$^3$) at the urban-street-network scale during August 2021 simulated by the IAQMS-street v2.0 (a–c), IAQMS-street v1.0 (d–f), and NAQPMS (g–i).

In this study, $O_3$ and $NO_x$ were observed on Beijing roads to evaluate the performance of IAQMS-streer v2.0 at the street-network scale during 12:00–18:00 h LT from 20 to 31 August 2021. As shown in Fig. 9, the distributions of the observed $O_3$ and $NO_x$ concentrations were reproduced by the coupled model, especially the low $NO_x$ concentrations on roads in urban areas (i.e., areas within the Fifth Ring Road). High $NO_x$ emissions, caused by increased traffic flow and a large proportion of trunks on roads, led to high $NO_x$ concentrations on roads in suburban areas. The $O_3$ concentrations in suburban areas were lower owing to the $NO_x$-$O_3$ titration mechanism. The statistical parameters between the observed and simulated data in three models at the street scale are shown in Fig. S1; the FAC2 values of $O_3$, NO, and $NO_2$ in IAQMS-street v2.0 reached 0.99, 0.42, and 0.83, respectively. In general, IAQMS-street v2.0 with dynamic traffic emissions can simulate the $O_3$ and $NO_x$ at the street scale efficiently.





**Figure 9.** Horizontal distributions of the observed O₃ concentration (a), NO concentration (b), and NO₂ concentration (c) at the street scale and simulation results by the two-way coupled model in IAQMS-street v2.0 (d–f), IAQMS-street v1.0 (g–i), and NAQPMS (j–l) (Imagery
© 2022 Google, map data © 2022 Google).



### 3.3 Contributions of local traffic emissions

We quantified also the contributions of local traffic emissions to the pollutant concentrations in Beijing. We conducted sensitivity analysis by comparing the results from IAQMS-street v2.0 and IAQMS-street v1.0 at regional scale. In IAQMS-street v2.0, the results were simulated using the two-way coupled model with dynamic emissions, with the mass transfer of pollutants from the street scale being calculated in the regional model; conversely, in IAQMS-street v1.0, the pollutant concentrations were simulated without considering the local traffic emissions at the regional scale. The equation for calculating the contributions of local traffic emissions to the pollutant concentrations in Beijing is the following:

$$P_{(x,y)} = \frac{(Conc_{(x,y,S1)} - Conc_{(x,y,S2)})}{Conc_{(x,y,S2)}} \times 100 \qquad (6)$$

where $(Conc_{(x,y,S1)}$ is average concentration of pollutants in grid location (x, y) simulated by IAQMS-street v2.0, $(Conc_{(x,y,S2)})$ is the mean concentration of pollutants in grid location (x, y) simulated by IAQMS-street v1.0, and $P_{(x,y)}$ is the contribution of local traffic emissions to grid location (x, y) in Beijing.

The distributed results from Eq. (3.1) are shown in Fig. 10. The contributions of local on-road vehicle emissions to the NO and $NO_2$ concentrations reached 90.63 and 82.66%, respectively, at the observation sites. The NO and $NO_2$ concentrations increased by 14.37 and 37.49 $\mu g/m^3$, respectively, due to the local traffic emissions, while the $O_3$ concentration decreased by 11.3%, indicating that urban areas in Beijing were in the VOC-sensitive region (i.e., the decreased $NO_x$ would increase the concentration of $O_3$). As shown in Fig. 10, local traffic emissions mainly influenced the concentrations of pollutants near urban ring roads and highways. The relative contributions of local traffic emissions to $NO_2$, NO, and $O_3$ reached 53.41, 57.45, and 8.49%, respectively. However, the contributions of local traffic emissions increased significantly with decreasing distance from main roads. The contribution of road vehicles to $NO_2$ reached as high as 93.5% on busy roads. Overall, local traffic emissions are important for the distribution of pollutants in Beijing.

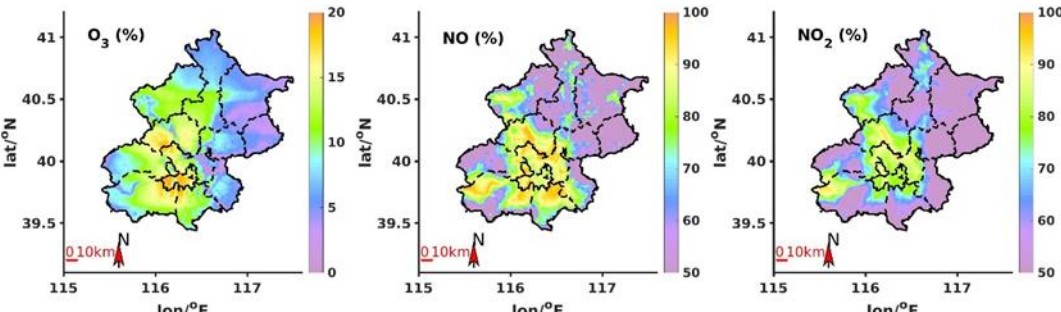

**Figure 10.** Contribution of local on-road vehicles to the $O_3$, NO, and $NO_2$ distributions in Beijing.

### 3.4 Evaluation of vehicle-control measures

To evaluate the influence of vehicle-control measures on the distributions of pollutants in Beijing, we conducted simulations based on two scenarios (i.e., S1_withoutLEZ and S1_upgrade, as described in Section 2.3) using IAQMS-street v2.0 and quantified the contributions of different control measures to the variations of pollutant concentrations. The on-road $NO_x$ emissions within urban areas in Beijing increased from 53.82 ton/day in S1 to 100.84 ton/day in S1_withoutLEZ, suggesting a decrease of $NO_x$ emissions by 46.6% due to the implementation of policy that establish the low emission areas, according to which vehicles below the China III emission standard were forbidden from entering urban areas. The on-road HC emissions were 23.81 ton/day in S1_withoutLEZ and 22.5 ton/day in S1. The influence of the control measures on the HC emissions was small because on-road HC was mainly emitted by LDVs (Fig. 4). A comparison of the spatial distributions of pollutants of S1_withoutLEZ and S1 is shown in Fig. 11. The $NO_x$ concentration decreased in urban areas, especially on busy roads such as the Fourth and Fifth Ring Roads. The NO and $NO_2$ concentrations decreased by 10 and 40 $\mu g/m^3$ on busy

roads, respectively. The decreased NO concentration caused $O_3$ accumulation. The maximum 8-h $O_3$ concentration in the
urban areas in S1 increased by 40 μg/m$^3$ compared to that in S1_withoutLEZ.

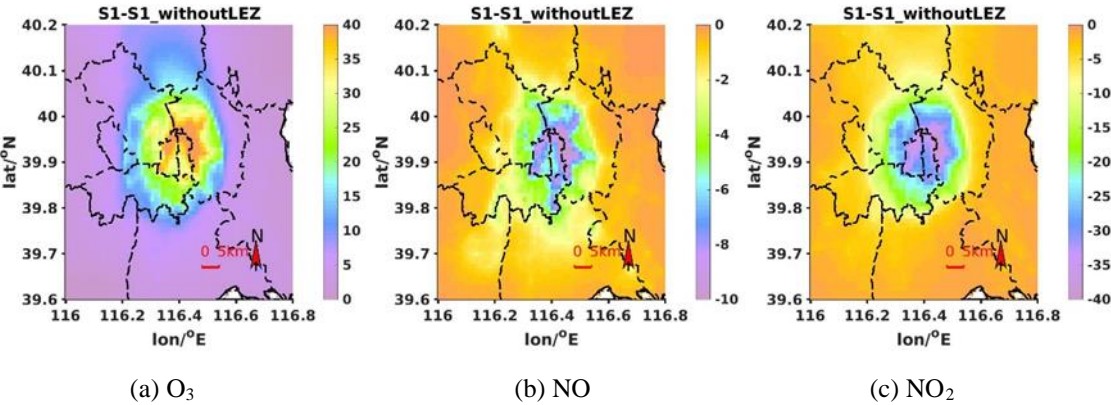

(a) $O_3$            (b) NO            (c) $NO_2$

**Figure 11.** Influence of policies that restrict vehicles in urban areas on the spatial distributions of (a) $O_3$, (b) NO, and (c) $NO_2$ in Beijing.

In the S1_upgrade scenario, on-road $NO_x$ emissions in urban areas decreased to 37.62 ton/day, i.e., they were 30.1% lower
than those in S1. The on-road HC emissions were 11.46 ton/day, i.e., they were 49% lower than those in S1. Results showed
that the upgraded vehicle-emission standards in S1_upgrade can significantly reduce traffic $NO_x$ emission and HC emission
in street network. The distributions of $O_3$, NO, and $NO_2$ simulated in S1_upgrade was compared to those in S1. As shown in
Fig. 12, the monthly average NO and $NO_2$ concentrations decreased from 0.52 and 15.45 μg/m$^3$ in S1 to 0.32 and 11.33
μg/m$^3$ in S1_upgrade. The decreased NO concentrations led to increased $O_3$ concentrations near road areas; however, in
areas away from roads (such as the southwest of Beijing, see Fig. 12), $O_3$ concentrations decreased due to the decreased $NO_2$
and HC emissions. The monthly average maximum 8-h $O_3$ concentration in urban areas eventually changed from 103.93
μg/m$^3$ in S1 to 103.08 μg/m$^3$ in S1_upgrade.

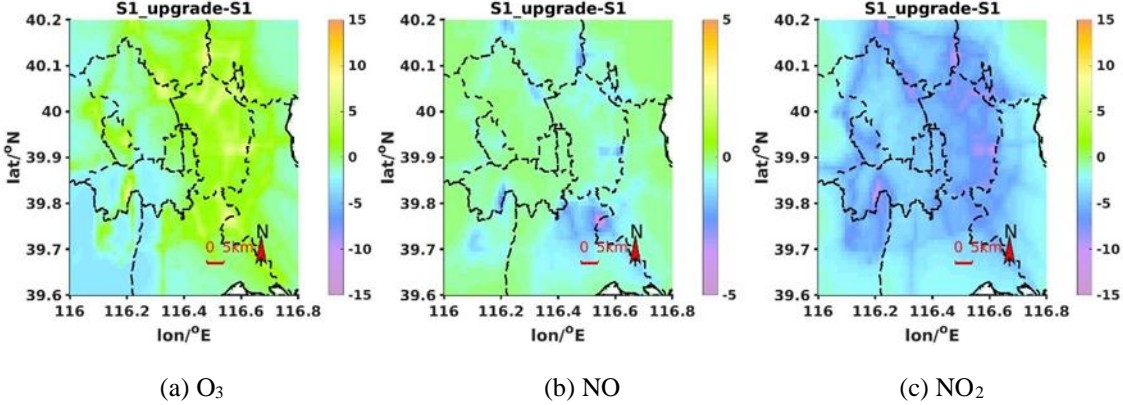

(a) $O_3$            (b) NO            (c) $NO_2$

**Figure 12.** Influence of policies with upgraded vehicle-emission standards on the spatial distribution of (a) $O_3$, (b) NO, and (c) $NO_2$ in
urban areas in Beijing.

A comparison of the simulated results showed that both the established low-emission zone and the upgraded traffic-emission
standard could effectively decrease the $NO_x$ concentrations near roads; however, when the road traffic emissions decreased,
the $O_3$ concentration near the road increased due to the decrease consumption of $O_3$ by NO. To decrease the $O_3$
concentrations in urban areas, controlling the HC and $NO_x$ emissions from other sources must be considered.



## 4 Conclusions

In this study, a new version of the regional-urban-street-network air-quality modeling system IAQMS-street v2.0 has been presented to simulate the urban background and street network pollution. A two-way coupled module was added in IAQMS-street v2.0 to transfer pollutants between regional and local street urban canopy, The influence of pollutants in street network

is considered in the concentration calculation on regional scale, which is not considered in previous version (IAQMS-street v1.0). Based on dynamic traffic emission inventories, we simulated the $O_3$ and $NO_x$ distribution characteristics at the regional and street scales. The simulation concentrations of $O_3$ and $NO_x$ in Beijing during August 2021 by the two-way coupled model (IAQMS-street v2.0), one-way coupled model (IAQMS-street v1.0), and regional model (NAQPMS) were compared. In addition, we quantified the contribution of local vehicle emissions to urban air quality and analyzed the influence of

traffic-control measures on the pollutant distributions. The simulation results of IAQMS-street v2.0 were improved, particularly at the street scale. The conclusions of this study are shown in below.

Dynamic emissions played an important role in the simulation of the coupled model. The HDT contribution to vehicle $NO_x$ emissions reached 34.4% and increased to 51.1% at midnight. $NO_x$ emissions were mainly distributed on the Fifth Ring Road and the highway outside of it owing to the implementation of traffic-control measures. By comparing the simulation

results of IAQMS-street v2.0 and NAQPMS at the monitoring sites, we found the $O_3$ underestimation and $NO_x$ overestimation were improved in IAQMS-street v2.0. And compared with IAQMS-street v1.0, due to the feedback of street model to regional model was considered, the spatial distribution of $O_3$ and $NO_x$ improved at regional scale.

The comparison between simulations and observations on roads showed that IAQMS-street v2.0 performed well at the street scale, and the FAC2 values between observed and simulated $O_3$, NO, and $NO_2$ reached 0.99, 0.42, and 0.83, respectively.

$NO_x$ concentrations on roads in urban areas were lower than those in suburban areas, which was confirmed by the observations, and the distributions of observed $O_3$ and $NO_x$ were reproduced by IAQMS-street v2.0.

The contribution of local traffic emissions to air quality is important in Beijing. The relative contributions of local traffic emissions on $NO_2$, NO, and $O_3$ reached 53.41, 57.45, and 8.49%, respectively; however, contributions increased significantly with decreasing distance from main roads, while the contribution of road vehicles to $NO_2$ reached 93.5% on busy roads. Both

the established low-emission zone and upgraded vehicle-emission standards could reduce the on-road $NO_x$ emissions; however, the $O_3$ concentration increases owing to the decrease consumption of $O_3$ by NO. To decrease the $O_3$ concentration in urban areas, controlling the HC and NOx emissions from other sources needs to be considered in future research.

**Code and data availability.** The source codes, observation data, and model output in our work are available online via Zenodo (https://doi.org/10.5281/zenodo.7298947, Wang and Li, 2022)

**Author contributions.** Tao Wang: Writing-original draft, Formal analysis, Methodology, Software. Jie Li: Supervision, Funding acquisition, Project administration, Writing – review & editing. Hang Liu: Data curation, Writing – review & editing, Formal analysis. Shuai Wang: Data curation. Youngseob Kim: Software, Writing – review & editing. Zifa Wang: Conceptualization, Methodology. Yele Sun: Data curation. Wenyi Yang: Writing – review & editing. Huiyun Du: Writing – review & editing. Zhe Wang: Formal analysis.

**Competing interests.** The contact author has declared that neither they nor their co-authors have any competing interests.

**Acknowledgements.** We thank the National Key Scientific and Technological Infrastructure project "Earth System Science Numerical Simulator Facility" (EarthLab). We acknowledge Xuemei Wang and Luolin wu from Jinan university for providing code and guidance on the use of ROE model.

**Financial support.** This work is funded by the National Key R&D Program of China (Grant No. 2022YFC3700703).





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
