# Peer review of "IAQMS-street v2.0: a two-way coupled regional-urban–street-network model system for Beijing, China"

_Geoscientific Model Development, 2023_

## Author Comment (AC1)

**Re: We thank the reviewers for your careful read and thoughtful comments on our manuscript. We have carefully taken your comments into considerations in preparing our revision, and below marked in blue is our response to your comments point by point, or you can see the revised manuscript for more details. Thanks again for your comments.**

**Specific:**

**Reviewer 1:** General comments: This study presents a new version of the regional urban road network air quality modelling system IAQMS-street v2.0 to simulate urban background and road network pollution. The manuscript is generally well organised, at this stage the reviewer has a positive attitude towards its publication. However, there are still some points regarding the numerical conditions that should be further explained and modified. The detailed comments of the reviewer are as follows:

Specific comments:

1. Line 163 "The fixed time step for interfacing between NAQPMS and MUNICH was 20 min, i.e., the same as that of the regional model." Is the time step 20 min also for MUNICH? As this study focuses on pollutant diffusion and chemical reaction at the street scale (100 m). The reviewer thinks that 20 min is too long. For example, the previous study on MUNICH with chemical reactions used a time step of 100 s. https://doi.org/10.5194/gmd-15-7371-2022. Please justify that the simulation results based on 20 min can achieve similar simulation accuracy with a smaller time step (e.g. 5 min). In addition, a sensitivity test should also be carried out on the time step for the interface between NAQPMS and MUNICH (e.g. 5 min).

Re: Thank you for reading this manuscript carefully and asking questions. Your comments are critical to improve the content of the manuscript. In MUNICH, the time step was setting as 1200s (20min) to corresponding with the time step in NAQPMS. In previous study, the performance of MUNICH with different time steps (100s and 600s) on street and background has been evaluated in Paris (Figure R1 and R2). In Figure R1, the results showed that the street concentrations of $NO_2$ and NO are numerically stable and independent of the choice of the time step in MUNICH with nonstationary approaches, and the nonstationary approaches decreased the influence on background concentration with different time step in Figure R2 (Lugon et al., 2020).

[Figure]

Fig R1. Daily average concentration of $NO_x$(a), $NO_2$(b), and NO(c) concentration (μg/m³) simulated by MUNICH in Paris with different time step (600s and 100s) using the stationary and nonstationary approaches ( Lugon et al., 2020).

[Figure]

Fig R2. Daily average concentration of NOx(a), NO$_2$(b), and NO(c) concentration (μg/m$^3$) simulated by SinG (a multi-scale model that couples MUNICH with the regional model Polair3D) in Paris with different time step (600s and 100s) using the stationary and nonstationary approaches (Lugon et al., 2020).

In this study, the nonstationary approaches have been used in MUNICH. Based on reviewer's comments, a sensitivity test has been carried out on the time step (Delta time=20min and Delta time=5min) for the interface between NAQPMS and MUNICH to quantified the influence of time step on simulation results. As shown in Fig R3 and R4, the simulation of NO$_x$ and O$_3$ with time step 20min and 5min were close to observations at monitoring sites. The hourly simulation results at monitoring sites were compared in Fig R5. The FAC2 between simulation results of O$_3$, NO, and NO$_2$ by time step 20min and 5min reached 0.99, 0.97 and 1.0 and The NMB of O$_3$, NO, and NO$_2$ is 0.03, 0.11, and 0.03. Overall, the simulation results based on 20 min can achieve similar simulation accuracy with a smaller time step (5 min). The sensitive analysis with different time step (20min and 5min) were added in the revised manuscript, please see the revised manuscript from line 370 to line 377 for more details.

[Figure]

Fig R3. Hourly variation of O$_3$, NO, and NO$_2$ concentrations (unit: μg/m3) during August 2021 at the Dongcheng-Tiantan (DC-TT) station and Xicheng-Wanshouxigong (XC-WSXG) station. Red lines indicate values simulated by IAQMS-street v2.0 with 20 min time step; blue lines indicate values simulated by IAQMS-street v2.0 with 5min time step; green lines indicate values simulated by the IAQMS-street v1.0; blue lines indicate values simulated by NAQPMS; and black lines indicate observations.

[Figure]

Fig R4. Observed and simulated average $O_3$, NO, and $NO_2$ concentrations during August 2021 from IAQMS-street v 2.0 with different time step (20 min: red points; 5min: blue points) at all pollutant monitoring stations in Beijing.

[Figure]

Fig R5. The comparison of simulated hourly $O_3$, NO, and $NO_2$ concentrations during August 2021 from IAQMS-street v 2.0 with different time step (20 min and 5 min) at all pollutant monitoring stations in Beijing.

2. Figure 1. How high is the bottom layer in this study? Please provide a reasonable explanation that the simulation results are not dependent on the height of the bottom layer.

Re: In this study, the height of the bottom layer in the regional model is 48.32m over the Beijing area, and the average building height in Beijing used in this study is 10.8m. As shown in Fig R6, more than 90% of buildings in Beijing have a height below 20m, and less than 4% of buildings exceeding 50m in height, which basic meets the requirement that the height of the street model need lower than the bottom height of the regional model in two-way coupled models (Lugon et al., 2020). Based on reviewer's comments, the information of building heights and the bottom layer height of regional model in Beijing were added in the revised manuscript. please see the revised manuscript from line 174 to line 177 for more details.

[Figure]

Fig R6. The frequency distribution of building height in Beijing urban area.

3. Figure 7. Why is the number of NO observations less than NO2 and O3? Also, it seems that only the daily averaged data are shown. The hourly averaged data should be shown as the prediction accuracy of the peak concentration is important.

Re: In Chinese National Ambient Air Quality Standards, $O_3$, $PM_{2.5}$, $PM_{10}$, CO, $SO_2$, and $NO_2$ are listed as six conventional pollutants for monitoring, and NO is not the main monitoring object, so the NO is not observed at all national monitoring sites, which caused the NO observations less than $NO_2$ and $O_3$, and the NO observation values are all integer digits in this study. Based on review's comment, the hourly averaged data is added in the revised manuscript (as shown in Fig R7), please see the revised manuscript from line 272 to line 273 and appendix figure (Fig. S1) for more details.

[Figure]

Fig R7. Observed and simulated hourly $O_3$, NO, and $NO_2$ concentrations during August 2021 from different models (IAQMS-street v2.0: red points; IAQMS-street v1.0: green points; NAQPMS: blue points) at all pollutant monitoring stations in Beijing.

Technical Corrections

4. IAQMS-street v2.0 could use more CPU time than IAQMS-street v1.0. It is helpful for the potential user of IAQMS-street v2.0 to know the detailed CPU time in this study for different scenarios.

Re: Based on review's comment, the calculation time of IAQMS-street v2.0 and IAQMS-street v1.0 are compared now. In this study, the NAQPMS used 4 nodes and 24 ppn (Processor Per Node) when MUNICH used 1 node and 28 ppn. During the study period, the calculation time is 121.6 h in IAQMS-street v2.0 and 96.2 h in IAQMS-street v1.0, and the calculation time increased to 212.8 h in IAQMS-street v2.0 with smaller time step (5 min). The detail description of calculation time for different scenarios has been added in the revised manuscript. Please see the manuscript from line 378 to line381 for more details.

Reference:

Lugon, L., Sartelet, K., Kim, Y., Vigneron, J., and Chretien, O.: Nonstationary modeling of NO2, NO and NOx in Paris using the Street-in-Grid model: coupling local and regional scales with a two-way dynamic approach, Atmos. Chem. Phys., 20, 7717-7740, 10.5194/acp-20-7717-2020, 2020.

Date of this revision:

28 Jun 2023

---

## Author Comment (AC2)

**Re: We thank the reviewers for your careful read and comments on our manuscript. We have carefully taken your comments into considerations in preparing our revision, and below marked in blue is our response to your comments point by point, or you can see the revised manuscript for more details. Thanks again for your comments.**

**Specific:**

**Reviewer 2:** In this manuscript, the authors designed a two-way coupled regional-urban–street network air quality model system, and applied and evaluated it in a megacity, Beijing, China. The topic is of great interesting to recognize the complex interactions of air pollutants between larger different scales in spatial dimension inducing by emissions, mass transform andchemisty among the scales. The manuscripts is generally well organized and the analysis is mostly sound. But some details need modify and some ambiguous presentation need clarify. I recommend a minor revision and my comments listed below.

Specific comments:

1. The title should include air quality, indicating you designed a two-way coupled air quality model system.

Re: Thank you for your comments. We appreciate the reviewer's positive evaluation of our work. Based on reviewer's comments, the title has been revised to: "IAQMS-street v2.0: a two-way coupled regional-urban–street-network air quality model system for Beijing, China". Please see the manuscript for more details.

2. In line 11 of the abstract, "gap" could be better replaced by "different" or other word implicating the large different between the concentration in reginal and street scales.

Re: The sentence has been revised to: "the concentrations of pollutants, such as ozone ($O_3$) and its precursors, have a large difference with the regional averages and their distributions cannot be captured accurately by traditional single-scale air-quality models". Please see the manuscript in line 11 for more details.

3. In line 27 and in the context, cannot say "O3 emissions".

Re: The sentence has been revised to: "The relative contributions of local traffic emissions to $NO_2$, NO, and $O_3$ concentrations were 53.41, 57.45, and 8.49%, respectively". Please see the manuscript from line 28 to line 29 for more details.

4. In line 121, dc-->dC;

Re: The "dc" has been revised to "dC", please see the manuscript in line 121 for more details.

5. In line 135, definition γ as mass transfer efficiency may better than mass flux.

Re: The sentence has been revised to: "γ is transfer efficiency between street and background concentration". Please see the manuscript in line 135 for more details.

6. In line 150, " in Eq. (3)," could be in Eq. (4)?

Re: The "Eq. (3)" has been revised to "Eq. (4)" in line 150. Please see the manuscript in line 150 for more details.

7. In figure 3b, I found a truck, but the "truck up" and "truck down" were both zero. Were the statistics (in red) right?

Re: In the object detection system, the identified vehicles were counted when they drives across the yellow line (as show in Fig 3b). In Fig 3b, the truck was not cross the yellow line so the "truck up" is keep to zero. Based on reviewer's comment, the Fig 3b is replaced to which truck has been counted (Fig R8). Please see the revised manuscript for more details.

[Figure]

Fig R8. (a) Locations of observation sites on different roads for vehicle information (Imagery © 2022 Google, map data © 2022 Google). (b) Detection results of vehicles on road by the YOLO system.

8. In figure 7, we can find the simulated NO and NO2 in regional (NAQPMS) higher than that in network simulations. So,what's the means of the presentation in abstract "the concentration of NOx at street scale is higher than that at the regional scale,"?

Re: In this study, the $NO_x$ concentration simulated by regional model NAQPMS and coupled model IAQMS-street were compared with observation data at regional scale (Fig 7), the results showed that the $NO_x$ concentration simulated by NAQPMS was overestimate. As mentioned in the abstract, "the concentration of $NO_x$ at street scale is higher than that at the regional scale" is try to compared $NO_x$ concentration in street-scale and regional-scale in the coupled model (the concentration of pollutants in street and in background). Based on reviewer's comment, the sentence has been revised to: "In the coupled model, the concentration of $NO_x$ at street scale is higher than that at the regional scale". Please see the manuscript from line 22 to line 23 for more details.

Date of this revision:

28 Jun 2023

---

## Author Response (AR2)

**Re: We thank the Editor and the Topic Editor's comments on our manuscript. We have carefully taken your comments into considerations in preparing our revision, and below marked in blue is our response to your comments point by point, or you can see the revised manuscript for more details. Thanks again for your comments.**

**Specific:**

**Editor ( Sarah Buchmann) :**

Your figures #3 and #9 are very low in quality and this is why, some parts are not even readable when zoomed in. For the next revision, please change these figures with higher quality versions.

Re: Thank you for your comment, the quality of figure #3 and figure #9 are improved now as show in below, and these figures have been updated in the revised manuscript.

[Figure]

Figure 3. (a) Locations of observation sites on different roads for vehicle information (Imagery © 2022 Google, map data © 2022 Google). (b) Detection results of vehicles on road by the YOLO system.

[Figure]

Figure 9. Horizontal distributions of the observed O3 concentration (a), NO concentration (b), and NO2 concentration (c) at the street scale and simulation results by the two-way coupled model in IAQMS-street v2.0 (d–f), IAQMS-street v1.0 (g–i), and NAQPMS (j–l) (Imagery © 2022 Google, map data © 2022 Google).

**Topic Editor (Leena Järvi) :**

1. Thank you for you good work concerning the reviewer comments. There are a few corrections still needed before I can accept the manuscript for publication. The language in the new section 2.5 needs to be improved: there are several mistakes so carefully check it through.

Re: Thank you for your comments. We appreciate the editor's positive evaluation of our work. Based on editor's comments, The section 2.5 has been revised to:

"In the base scenario (S1), the time step was set as 20 min in IAQMS-street v2.0. An additional simulation scenario was set with a time step of 5 min in IAQMS-street v2.0 to analyze the influence of the time step in the coupled model. The comparison of simulated pollutants in IAQMS-streetv2.0 with different time

steps is shown in Fig. S3 and Fig. S4. The FAC2 between simulation results of $O_3$, NO, and $NO_2$ for the time step of 20 min and 5 min reached 0.99, 0.97, and 1.0. The NMB of $O_3$, NO, and $NO_2$ is 0.03, 0.11, and 0.03. Overall, the simulation results based on a 20-min time step can achieve similar simulation accuracy with a smaller time step of 5 min. The results showed that the simulated pollutants are numerically stable in the coupled model with nonstationary approaches, which is consistent with the previous research findings (Lugon et al., 2020).

In terms of computational time, the NAQPMS used 4 nodes and 24 ppn (Processor Per Node) while MUNICH used 1 node and 28 ppn in this study. During the study period, the calculation time was 121.6 h in IAQMS-street v2.0 and 96.2 h in IAQMS-street v1.0, and the calculation time increased to 212.8 h in IAQMS-street v2.0 with a smaller time step of 5 min, which means that the smaller the time step, the longer the computation time.".

Please see the revised manuscript for more details.

2. L198-199: The reference YOLOv5s should be updated so that the github link and date of access are in the reference list and not in the main manuscript following the editorial guidelines.

Re: Based on the editorial guidelines, the reference to YOLOv5s has been updated, and the Zenodo link and date of access are included in the reference list as follows: "Jocher, G.: YOLOv5 by Ultralytics (Version 7.0), Zenodo [Code], https://doi.org/10.5281/zenodo.3908559, 2020, last access: 13 August 2023". Please see the revised manuscript for more details.

3. L421: Similarly the Zenodo link to the dataset Wang and Li, 2022 should be in the reference list as Tao Wang, Jie Li, & Zifa Wang. (2022). IAQMS-street online model data. https://doi.org/10.5281/zenodo.7298948

Re: The Zenodo link of the dataset used in this study has been added in the reference list as follows: "Wang, T., Li, J., and Wang, Z. F.: IAQMS-street online model data, Zenodo [data set], https://doi.org/10.5281/zenodo.7298948, 2022.". Please see the revised manuscript for more details.

4. There are multiple mistakes in the reference list with paper names in capital letters, references Ministry of Ecological Environment of China. are not referred in the main text etc. Please check all references carefully.

Re: The reference list has been checked now and the references of Ministry of Ecological Environment of China have been removed from the reference list. Please see the revised manuscript for more details.

Date of this revision:

13 Aug 2023

---

## Author Response (AR3)

**Re: We thank the Editor's comments on our manuscript. We have carefully taken your comments into considerations in preparing our revision, and below marked in blue is our response to your comment, or you can see the revised manuscript for more details. Thanks again for your comments.**

**Specific:**

**Editor:**

Your figures #5 and #9 are very low in quality and this is why, some parts are not even readable when zoomed in. For the next revision, please change these figures with higher quality versions.

Re: Thank you for your comment, the quality of figure #5 and figure #9 are improved now as show in below, and these figures have been updated in the revised manuscript.pdf, higher quality figures can be found in the Figures.zip or manuscript.docx.

[Figure]

Figure 5. Horizontal distributions of the NOx and HC emissions (unit: kg/km/day) at the street-network in Beijing urban and suburban area during August 2021 (Imagery © 2022 Google, map data © 2022 Google).

[Figure]

Figure 9. Horizontal distributions of the observed O3 concentration (a), NO concentration (b), and NO2 concentration (c) at the street scale and simulation results by the two-way coupled model in IAQMS-street v2.0 (d–f), IAQMS-street v1.0 (g–i), and NAQPMS (j–l) (Imagery © 2022 Google, map data © 2022 Google).

Date of this revision:

1 Sep 2023